# Emotional Suffering After the COVID-19 Pandemic: Grieving the Loss of Family Members in Brazil

**DOI:** 10.3390/ijerph21111398

**Published:** 2024-10-23

**Authors:** Pamela Perina Braz Sola, Manoel Antônio Santos, Érika Arantes Oliveira-Cardoso

**Affiliations:** Faculty of Philosophy, Sciences and Letters of Ribeirão Preto, University of São Paulo, Ribeirão Preto 14040-900, São Paulo, Brazil; masantos@ffclrp.usp.br (M.A.S.); erikaao@ffclrp.usp.br (É.A.O.-C.)

**Keywords:** bereavement, pandemics, COVID-19, death, family

## Abstract

(1) Background: Brazil has been severely affected by the COVID-19 pandemic, with over 700,000 deaths and, consequently, a drastic increase in the number of bereaved individuals. This study aims to understand the emotional suffering after the COVID-19 pandemic in Brazilian adults whose family members have died due to COVID-19. (2) Methods: A clinical–qualitative, cross-sectional, descriptive–exploratory study with a sample composed of 10 bereaved family members was used. Data collection took place in July 2021 through individual semi-structured interviews conducted via video call. The interviews were fully transcribed and subjected to thematic analysis. The corpus was analyzed based on Parkes’ theory of mourning, in dialog with research conducted in the pandemic context. (3) Results: The results were organized into three categories: Living the anticipation of loss in an unknown world; Living through grief in a changed world; and Glimpsing a new possibility of living. (4) Conclusions: The rupture of the presumed world in times of the pandemic, the impossibility of bidding farewell to deceased loved ones, and low levels of social support hindered the process of mourning during the health crisis.

## 1. Introduction

The World Health Organization (WHO) declared the COVID-19 pandemic on 11 March 2020 [1]. Just over three years later, on 5 May 2023, the WHO declared the end of the public health emergency of international concern, indicating that COVID-19 would be managed by countries with a status similar to that of other infectious diseases, no longer following emergency measures [2]. Brazil was one of the countries most drastically affected by a lack of control over the new coronavirus (SARS-CoV-2). The virus spread quickly, with distinct temporal and geographic spread patterns [3]. Studies show severe flaws in the implementation, coordination, and financing, with unequal responses to combat the pandemic nationally [4,5,6,7]. Thus, different Brazilian regions were seriously affected at different times, with peaks of contamination and high morbidity and mortality rates recorded throughout 2020 and 2021 [8].

An analysis of the rates of confirmed cases and deaths shows that Brazil’s first wave of the pandemic occurred from February to November 2020. The significant increase in the number of positive cases and deaths in November initially detected in Manaus, AM, indicated the beginning of the second wave with its peak in March 2021 [9,10,11]. The number of deaths in April of the same year exceeded the total number recorded throughout 2020, reaching 400,000 deaths [12,13]. From the pandemic’s beginning until 5 May 2023, when the WHO officially revoked the global emergency, Brazil recorded 701,494 deaths due to COVID-19 [14].

The Ministry of Health’s chaotic management marked the pandemic in Brazil; poor leadership impeded a crisis committee from being established, hindering coordinated communication between states and cities [15]. Furthermore, the controversial practices the government adopted, endorsing and disseminating fake news, discouraged the population from wearing face masks and adhering to vaccination, hindering compliance with the distancing measures intended to minimize the severity of the pandemic and to promote Brazilian’s health and well-being [16,17,18]. Mass immunization in Brazil began in January 2021 [13]. Even with the support of the National Immunization Program, one of the largest and most complete vaccination programs in the world, Brazil presented low population vaccination rates; there was much delay in purchasing and provisioning vaccines [15,19]. The estimation is that 95,500 deaths caused by COVID-19 could have been avoided if there had not been a delay in acquiring the vaccine [20,21].

In Brazil, the thousands of deaths caused by COVID-19 resulted in thousands of mourners. The exceptional situation imposed by the pandemic dramatically impacted the experience of terminal illnesses and grief, which were profoundly affected by hitherto unprecedented factors [22,23]. Thus, the experience of mourning during the pandemic involved multiple deaths in the same family, families being distant from patients on the verge of death, being unprepared for death, and the impossibility of farewell rituals; funerals and social support were prohibited or shortened [24,25,26,27,28,29,30]. In the Brazilian context, paying last respects to a loved one is a deeply cultural act, not only providing an opportunity to honor and say farewell to the deceased but also to receive support from the community. When these rituals cannot be performed, mourners often report feeling that the grieving process is left incomplete, and the inability to share their pain intensifies their sense of loss [28].

A meta-synthesis was conducted with the objective of summarizing and reinterpreting the results of qualitative studies on the experience of losing family members during the pandemic by a thematic synthesis. It indicated that experiences of loss in this context were negatively impacted by the imperatives of physical distance, restriction of hospital visits, technology-mediated communication, and the prohibition or restriction of funerals. These changes resulted in experiences marked by feelings of loneliness and helplessness, which should be considered when planning intervention strategies that favor communication between family members with the afflicted loved one and with the healthcare team, enabling welcoming and creating alternatives for farewell rituals [29].

Therefore, the COVID-19 pandemic, an unprecedented global event, caused sudden changes in the daily lives of people who experienced the constant threat of death, either of themselves or loved ones, changing prospects and leading to a feeling of insecurity and unpredictability [31,32]. The world as it was known thus far radically changed. Parkes (2010) [33] names this radical change in world perception as “changing the assumptive world”, which essentially occurs during the grieving process; dealing with a significant loss requires one to adapt to changes and rebuild another way of life [28,32,33].

Given these particularities, this article, by exploring grief in Brazil during the COVID-19 pandemic, highlights how extreme conditions—high mortality, misinformation, the collapse of the healthcare system, economic crisis, and social divisions–shaped a unique grieving experience with challenges and complexities distinct from those in other parts of the world. This study has the potential to contribute to the literature on grief in extreme crisis settings and social inequality. Understanding the problems caused by the pandemic in the grieving processes and considering the prolonged crisis might contribute to the care provided to bereaved people and design strategies to cope with future pandemics by considering the needs of grieving populations. Bereavement persists long after death; thus, research remains relevant and necessary, especially in the Brazilian context, where there is a scarcity of qualitative research published in widely circulated journals [29]. For this reason, this study aims to understand the emotional suffering after the COVID-19 pandemic in Brazilian adults whose family members have died due to COVID-19.

## 2. Materials and Methods

### 2.1. Study Design

This is a clinical–qualitative, cross-sectional, descriptive–exploratory study. The clinical–qualitative method consists of the particularization of qualitative methods for describing and interpreting meanings attributed to experiences in health contexts [34,35].

### 2.2. Study Setting and Participants

The sample was selected using the snowball non-probability sampling technique, in which an invitation to participate in the study was published on the research laboratory’s social network. The snowball sampling method is widely used in qualitative research, where the researcher’s focus is not on numerical representativity, and there is no concern with random selection. The goal is to recruit participants who can offer privileged information on the topic. The focus is not on controlling variables to increase the generalizability of the results but rather on gaining an in-depth understanding of a phenomenon from the perspective of the participants. On one hand, an advantage of this method is that it fosters an environment of trust, as the participants are contacted through the recommendation of someone they know. On the other hand, one disadvantage is the risk of limited diversity in the sample, as participants often share similar characteristics (a phenomenon known as ‘anchoring’). To mitigate these potential biases, certain precautions were taken in selecting participants, following guidelines from the literature [36], such as ensuring diversity among the initial participants, not limiting data collection to a single dissemination of the study, and promoting the research through various channels and contexts.

The sample size was established according to the theoretical saturation criterion, i.e., data collection ceases when the researcher considers that the responses provided by new informants are significantly repetitive, with no substantially new content, considering the topics already covered and the responses obtained [34]. The following inclusion criteria were adopted: having experienced the loss of a family member in the last 12 months due to COVID-19, being 18 years old or older, living in Brazil, and having access to the Internet. People presenting significant comprehension difficulties or limitations to use the Internet and the online communication platform, which would prevent people from engaging in the interview, were not selected. Only one participant was excluded due to difficulty accessing the online data collection platform. The convenience sample comprised ten family members of COVID-19 victims (Table 1).

A brief individual description of the participants (fictitious names):
Jessica, 35 years old, freelancer, lost her mother (64 years old).Antonia, 48 years old, salesperson, lost her mother (70 years old), her father (85 years old), and her brother (44 years old).Romeo, 23 years old, historian, lost his mother (61 years old).Rebecca, 31 years old, business administrator, lost her husband (30 years old).Costa, 52 years old, driver, lost his wife (46 years old).Reissa, 45 years old, unemployed, lost her husband (44 years old), her mother-in-law (70 years old), and her father-in-law (85 years old).Larissa, 30 years old, manicurist, lost her mother (66 years old).Rita, 21 years old, receiving clerk, lost her father (59 years old) and her grandmother (80 years old).Liz, 31 years old, public servant, lost her father (51 years old) and her grandfather (72 years old).Regina, 22 years old, student, lost her grandfather (74 years old).

### 2.3. Data Collection

A script was developed to guide the semi-structured interviews. The questions were based on the review of scientific studies on the topic [29], complemented by the researchers’ experience. This review highlighted the importance of understanding the pre-death experiences and post-death experiences of these family members. In this sense, the script began with experiences before COVID-19, followed by the impact of the diagnosis, the experience of treatment, the communication of the death, and life after the loss. The semi-structured interview was chosen to support this clinical–qualitative research, as it enables both researchers and participants to be autonomous and free to dialog and address the various aspects of this topic. Thus, the interaction between interviewer and interviewee favors more spontaneous responses, the exploration of more complex subjects, and the collection of in-depth data [34]. Examples of questions asked include Can you share about your relationship with the family member before their illness? How was the diagnosis communicated? What significant experiences did you have during the treatment? How was the news of the death conveyed? Was there a burial? How are you feeling at the moment? Do you have any plans for the future? All interviews were held online through a video conferencing platform, which can be accessed via computer or mobile phone. Instructions on using the platform and ensuring confidentiality were shared with participants in advance. The individual online interviews were performed in July 2021 in a single meeting that lasted approximately one hour.

### 2.4. Data Analysis

The interviews were audio and video recorded after consent; however, only the audio was used for analysis. They were transcribed verbatim, respecting the sequence and the statements as they occurred. Following the reflexive thematic analysis technique proposed by Braun and Clarke (2019) [37], repeated patterns of meanings identified in the data obtained through interviews were analyzed and reported. This procedure can be applied across a range of theoretical and epistemological approaches and provides a flexible and helpful research tool for a detailed analysis of complex data. The aim was to conduct an inductive, semantic, and realistic data analysis.

Regarding the analysis procedure, three researchers independently conducted thorough readings of each interview and identified themes. To organize the material and systematize the analysis process, QDA Miner Lite software (version 2.0.7) was used, which strengthens methodological rigor and allows for different types of analysis. The researchers imported the interview material into the QDA software, which provided a range of tools for editing and grouping the analyses. After conducting thorough readings, the researchers were able to create and organize the codes, exporting excerpts into spreadsheets that formed the categories and/or subcategories. The evaluators discussed their lists, cross-referencing the data from the analysis of each interview and recording the convergences and divergences found. These were examined, discussed, and validated by consensus, resulting in a single list of themes. Subsequently, the results were discussed with other members of the research group to refine and validate the thematic categories developed, as proposed by Yardley (2000) [38]. The themes were identified during the coding process based on the data collected instead of categories or prior knowledge. Furthermore, analysis of the semantic content, in which meanings were identified according to the analysis of the participants’ statements, first involved the description of data and, subsequently, interpretation (which involves the theorization of patterns and their meanings) and, finally, an approach to the literature on the topic.

The theoretical framework of this study is based on the theory of grief proposed by Parkes (2010) [33]. According to this author, everyone constructs a worldview that involves habits and thoughts, which need to be revised after significant changes, a process he calls Psychosocial Transition. Losses create discrepancies between the internal world and the world that comes into existence, turning a once familiar world into something suddenly unknown. According to the theory, some events are so impactful that they can invalidate entire areas of our assumed world, leading individuals to experience the feeling that nothing is certain anymore, making it very difficult to process these significant changes. The events that pose the greatest challenge for adaptation are those that require a reassessment of one’s worldview, involve lasting (rather than temporary) changes, and occur within a short period of time. In this context of multiple losses, as was the case during the COVID-19 pandemic, unexpected deaths combined with various other forms of grief can undermine the assumption that the world is a safe place. Changes face resistance, as individuals cling to their old model of the world, and in many cases, support networks and protective measures are necessary to aid the individual during this transition process.

As a complement to Parkes’ theory, we applied the concept of overload from the dual process model, recognizing that in the context of numerous changes and the need for readaptation, the grieving person may feel overwhelmed by more issues than they can manage. In this scenario, the coping process, which involves oscillating between loss-oriented and restoration-oriented stressors, may not take place. This is because the burden of one or both of these poles, or even additional stressors unrelated to the grief itself, can become too intense and overwhelming [39].

### 2.5. Statistical Analysis

No formal statistical hypothesis testing or sample size calculation was performed; all reported data were qualitative, and descriptive and categorical variables were reported as counts (frequencies and percentages). The sample size of 10 family members was considered reasonable to allow for the identification of key themes in the results. As a clinical–qualitative study, there was no intention to generalize the findings.

### 2.6. Ethical Aspects

The Institutional Review Board approved this study, which followed ethical guidelines concerning volunteers, by Resolution No. 466/12 on research involving human subjects [40]. Furthermore, to ensure the principles of beneficence, non-maleficence, and justice, a free online therapeutic group was offered to the participants whose need for continued care was identified [41].

The online interviews were conducted during a time when the Federal Council of Psychology had established guidelines for psychological care mediated by technology, and these instructions were used to guide the interviews. At the end of the interviews, participants were asked how they experienced the contact, and all affirmed that it was a moment in which they felt welcomed and had a safe space to express their pain. Furthermore, to uphold the principles of beneficence, non-maleficence, and justice, a free online therapeutic group was offered to participants whose need for continued care was identified [41,42]. Nine of the ten interviewees participated in the group, with only one family member declining the offer [41,42].

The study’s objectives and the conditions concerning professional confidentiality were informed to each participant, and only those who agreed with the terms and signed a free and informed consent form were included. Considering the context of an online interview, the signing of the form was replaced by acceptance expressed at the interview, as instructed in Circular Letter No. 2/2021/CONEP/SECNS/MS [43]. Thus, the participants’ acceptance was recorded via audio and video, and a copy of the form signed by the researchers was immediately sent to the participants. Fictitious names were used to identify the participants.

### 2.7. Reflectivity/Positionality Statement

To reduce the influence of the researchers’ perspective, the research questions and prompts were crafted to be as unbiased and open-ended as possible to prevent “leading the participant” with the researchers’ views. Reflexivity practices were also implemented, including debriefing sessions following the initial interviews. These practices involved providing detailed descriptions of the study participants and context, using direct quotes to support the findings, and thoroughly documenting the research process. In the debriefing session, the researchers reviewed and discussed the analysis process and results among themselves and with the participants to ensure clarity and understanding of the data collected.

## 3. Results

Three themes were developed, including six sub-themes, as shown in Table 2.

### 3.1. Living the Anticipation of Loss in an Unknown World

This theme was composed of two sub-themes, which are presented in Table 3 with illustrations of patients’ reports.

#### 3.1.1. Transformations in Family Daily Life

Monitoring potential contamination, illness, and the sudden death of family members during the critical moments of the pandemic involved drastic changes in the occupation of domestic spaces where families typically gathered. For example, individuals could no longer engage in familiar activities such as lying on a parent’s lap to watch TV in the living room, as family members had to isolate themselves upon showing symptoms of illness. Caregivers found themselves unable to visit loved ones, leading to prolonged separations. In some cases, physical barriers were created in attempts to maintain safety while also providing emotional support.

These disruptions in daily family life became even more challenging after the death of a family member. Individuals reported having to navigate new boundaries, such as creating distance to avoid contagion when seeking comfort from grieving family members. The absence of familial support was keenly felt, particularly in larger families that traditionally emphasized unity. The experience of isolation during these times did not alleviate the grieving process; rather, it exacerbated feelings of helplessness and intensified the state of shock following the loss.

#### 3.1.2. Changes in Healthcare Settings

The barriers preventing families from entering hospitals and the requirement to remain isolated at home led to feelings of despair and helplessness among many individuals. Romeo was the only participant allowed to accompany his elderly mother during her hospital stay. In contrast, others who obtained authorization to visit their relatives described their experiences as driven by intense desperation. For instance, Rebecca managed to secure a few visits to her hospitalized husband, which she viewed as a “privilege”, only after expressing her distress at the hospital reception. Larissa, feeling terrified, received authorization to see her mother in the ICU, an exception granted after she pleaded with a doctor. Reissa, on the other hand, was not permitted to visit her husband in the hospital and could only see him briefly outside the hospital while he was being transferred to another city, again after begging a doctor.

These extreme situations illustrate how despair and helplessness intensified due to the exceptional context imposed by the pandemic, which necessitated drastic changes in care protocols and increased feelings of insecurity and uncertainty experienced during the hospitalization of family members who died within a short time. Moreover, the deterioration of family members’ health often hindered communication and the sharing of more intimate and emotional content. For example, Rita managed to see her father once via video call, but his weak voice and exhaustion limited their interaction. Consequently, she was left uncertain about his emotional state, unable to ascertain whether he felt fear, sadness, or hope. Rebecca described her last conversation with her husband via video call as “horrible,” expressing a deep desire to have been physically present in the ICU, even if only to hold his hand, say something comforting, or say goodbye before he was intubated. Similarly, Larissa felt regret for not being able to share a final hug with her mother.

### 3.2. Living Through Grief in a Changed World

This theme was composed of two sub-themes, which are presented in Table 4 with illustrations of patients’ reports.

#### 3.2.1. Sudden Losses: Reality Took on Unreal Characteristics

Amid unusual and accelerated events, the illness and death of family members were perceived as a nightmare, characterized by a sense of unreality. Participants described how their reality took on surreal qualities; for example, Rebecca reflected on how just months before her husband’s death, they were making future plans together, which contrasted sharply with the suddenness of his passing. Rita noted that the sequence of events unfolded quickly and painfully, with the rapid progression of illness leading to the death of loved ones within days or weeks, intensifying their suffering. Rebecca further emphasized the abrupt nature of her husband’s decline, noting that he seemed perfectly healthy one moment, playing with their son, only to be gone shortly after.

The speed at which illness escalated was a common theme among participants. Romeu described coping by taking “one day at a time”, acknowledging that his mother’s condition could change drastically overnight. Similarly, Larissa recounted the sudden worsening of her mother’s health, stating that COVID-19 presented a unique challenge. Unlike cancer, where there is often a terminal phase allowing for preparation and closeness, COVID-19 could lead to unexpected complications like low oxygen levels or cardiac arrest within moments. Romeu echoed this sentiment, indicating that had he been given time to understand his mother’s deteriorating condition, he could have mentally prepared for her death. Feelings of helplessness were evident as well. Liz experienced her father’s death as a shocking and sudden loss, making it difficult for her to accept the reality of the situation. Antonia described how her life was abruptly disrupted, highlighting the emotional upheaval caused by the unexpected nature of death.

Following the initial numbness that accompanied the news of a loved one’s passing, many participants expressed that it took time to process their grief. Rita shared that she still felt as though she had not fully realized her loss and was still waiting for the reality of her situation to sink in. In a period marked by numerous deaths, individuals like Reissa, Liz, Antonia, and Rita, who faced the simultaneous illness of multiple family members, noted that the rapid pace of events coincided with the clinical deterioration of their loved ones. Reissa described her overwhelming concerns for her mother-in-law, father-in-law, and husband, all of whom faced health crises at the same time. The loss of a unique and significant family member, such as Larissa’s mother, occurred alongside many other deaths, contributing to a climate of political indignation and complicating the grieving process. Jessica conveyed that she struggled to feel a positive sense of longing, instead experiencing anger when remembering her loved ones.

#### 3.2.2. Changes in Rituals and Accelerated Farewells

Participants noted that changes to burial practices during the pandemic, particularly the prohibition of viewing the bodies of deceased family members, led to significant distress. Families were often forced to accept sealed coffins, which heightened their concerns and fantasies regarding the state of the bodies, which were buried in black bags. This lack of access to the deceased deprived families of the opportunity to pay their last respects, ultimately diminishing the meaningfulness of the burials. As a result, funerals were reduced to their most practical and instrumental functions, serving primarily to provide a final resting place for the body without the accompanying ceremonies that typically provide closure.

For those who were allowed to have open-casket funerals, such as Reissa, viewing the deceased’s body facilitated a deeper understanding of the finality of death. Participants expressed that seeing their loved ones helped them grapple with the reality of their loss. Rita, for instance, conveyed that witnessing her father’s body would be essential for her to accept his death, a sentiment echoed by Liz, who compared their need to see the body to that of Saint Thomas, emphasizing that visual confirmation was necessary for belief in the reality of death. Conversely, the inability to organize a proper funeral made it challenging for families to comprehend and accept the finality of their loss. Larissa described the painful experience of a simple burial without a wake, reflecting how this absence led some family members to cling to the notion that their loved ones were still alive, as they had not seen them in their coffins. Liz also recognized that not being able to say goodbye to her father’s body complicated her acceptance of his death.

Moreover, participants reported that when burials and funerals were permitted, the ceremonies were brief, leaving little time for family members to comprehend and bid farewell. Antonia illustrated this by sharing that, within two hours of receiving the news of her loved one’s passing, they had to proceed with the burial without any formal wake. The urgency surrounding these events left families feeling deprived of the opportunity to create lasting memories and meaningful farewells. The experience of losing family members to COVID-19 was profoundly affected by the pandemic’s inherent factors, particularly the restrictions on social gatherings in spaces that were previously communal. The inability to gather significant people for wakes or at home contributed to the mourning process occurring in isolation, stripping it of its shared meaning. This resulted in a death that felt distant and challenging to comprehend, followed by grief that lacked the necessary space for expression and sharing.

### 3.3. Glimpsing a New Possibility of Living

This theme was composed of two sub-themes, which are presented in Table 5 with illustrations of patients’ reports.

#### 3.3.1. Vaccine: Anger at the Delay and Hope for the Vaccination Itself

In addition to experiencing a combination of extreme situations, many individuals felt a mix of loneliness, sadness, and anger due to the lack of vaccination opportunities for their family members. This sense of indignation was particularly pronounced as they recognized that the deaths of their loved ones could have been prevented if vaccines had been distributed earlier and more efficiently.

The inability to prevent family members from contracting COVID-19 before the vaccine became available fostered a pervasive feeling of hopelessness. Many individuals expressed sorrow upon realizing that their loved ones succumbed to a disease that could have been avoided with timely vaccination. Mental health challenges were common, particularly for those who had taken significant precautions to protect themselves, only to see family members fall ill and die from the virus.

Additionally, the timing of deaths was particularly tragic for those who lost loved ones shortly before they would have been eligible for vaccination, further compounding their grief. The reflections of those affected revealed painful memories and considerations about how life might have unfolded differently had vaccines been distributed more promptly. Despite the overwhelming grief, there were glimpses of hope, as many individuals had already received their first vaccine dose or were about to do so at the time of the interviews.

This vaccination experience elicited complex emotions, as receiving the vaccine served as a stark reminder of the losses endured due to delays in vaccination efforts. Many felt a sense of anger and betrayal, believing that their loved ones’ deaths were exacerbated by governmental negligence regarding the timely rollout of the vaccine, which intensified their mourning experience.

#### 3.3.2. Catching a Glimpse of a New Way of Living

Many individuals expressed a profound lack of perspective and difficulties in envisioning a future without their deceased family members. The death of loved ones had an enduring emotional impact, leaving some individuals feeling as though they were still living in despair, doubt, and anguish, grappling with uncertainties about their lives moving forward. Feelings of being completely lost and desperate were common, especially in the wake of sudden and traumatic losses.

The cemetery, a symbolic space where the bodies of family members are interred, transformed into a poignant reminder of the irrevocable loss experienced. For many, returning to the cemetery became a painful confrontation with their grief. Some individuals reported a lack of courage to enter the cemetery, with even the act of approaching its walls triggering painful memories of their loved ones’ deaths. The traumatic nature of their experiences was evident, as the memory of the circumstances surrounding these losses remained vivid and distressing.

Communication related to hospitalizations also left lasting emotional scars. Updates from ICU doctors, often received through messaging apps, led to heightened anxiety, with the sound of phone notifications now eliciting racing hearts and fear. Nighttime panic attacks, characterized by tightness in the chest, were also reported, emphasizing the long-term psychological effects of these traumatic experiences. The sentiment that “nothing will ever be the same again” captured the profound impact of these losses.

In contrast, with the prospect of mass vaccination on the horizon, individuals began to reestablish face-to-face contact with family members. They started planning trips to reunite with loved ones and sought new ways to process their grief. The experience of being vaccinated brought about a shift in perspective, leading to renewed hope for the future and a sense of change. Individuals felt a sense of protection, described as a “little shield”, while also recognizing the importance of remaining cautious. The vaccination experience fostered a sense of freedom, allowing some to alternate between days of suffering and moments of well-being, ultimately reinvigorating their investment in future possibilities.

## 4. Discussion

This research was conducted in the Brazilian context, at the beginning of 2021, a time and place where guidelines on physical isolation or social distancing were not properly communicated. The chaotic management of the pandemic in Brazil endorsed denialist narratives, spread fake news, discouraged the use of masks, contradicted recommendations for physical distancing, and undermined the national immunization plan. As a result, the country had one of the highest global death rates, including thousands of avoidable losses [4,16,17,18,21,27]. The ten participants had resumed routines close to normal, relaxing mask usage, continuing in-person work, and attending larger social events, even during the period of highest contamination rates. With the deaths of family members, there was an intensification of the fear of contagion and an increased perception of risk, leading to a period of greater isolation. In July 2021, when the interviews were conducted, vaccination had already started nationwide, contributing to a broader outlook on the future and sparking some, albeit cautious, hope of reuniting with family, resuming life plans, and taking meaningful trips.

Considering Parkes’ assertion that certain events are so impactful that they invalidate entire areas of the presumed world, making it difficult to process significant changes and intensifying attachment to the known world, it is possible to understand, albeit incompletely, this denial of the severity of the situation and the choice to believe in fake news. By denying the new reality, individuals protected themselves from the challenges of the external world, avoiding situations where the discrepancies between their internal and external worlds would become evident. However, this denial was broken by the illness and death of a family member, which “forced” them to deal with yet another need for major adaptation.

As proposed by Parkes (2010), the loss of a family member can be understood as a complex and multifaceted phenomenon [33]. Thus, like a shock wave, the death of a family member spreads throughout the family, causing an immediate impact in the medium and long term. Bereavement is defined as a process of psychosocial transition in which it is difficult to precisely pinpoint what has been lost when a loved one dies. Hence, it is vital to consider the bond established between the bereaved person and the deceased to understand the roles each played and how the loss is experienced after death [44]. Therefore, one has to pay attention to individuals and their familial and cultural contexts to understand the diversity and complexity of loss processes. Additionally, one must consider the factual circumstances of death and its specific meaning for the family [33,43,44]. Parkes [45] and Walsh and McGoldrick (1991) [44] acknowledge that multiple, premature, and sudden deaths add stressors to the grieving process, which, in turn, depends on the stage of one’s life cycle and respective contextual, social, and cultural conditions. The two thematic categories previously introduced were created based on the premise that “A death occurs at a particular time and place”. [33] (p. 9).

Due to the high risk of contamination and the hospitals’ overload, most participants were not allowed to visit their hospitalized family members. Such an impediment was identified by other studies conducted during the pandemic, in which bereaved family members reported intense distress, despair, and helplessness resulting from the isolation and the impossibility of making hospital visits [29,44,46]. Due to the closure of hospital spaces, studies encouraged the use of resources mediated by digital information and communication technologies to enable communication with hospitalized family members in an online environment, using video calls or the exchange of audio and text messages [46,47]. Given the extraordinary circumstances, not being able to hug a loved one acquires an unusual emotional dimension, signifying a lack of physical and emotional closeness in such a delicate time. This situation persisted after the death of family members.

According to Parkes [45] (p. 159), “all grief is traumatic, but some grief is more traumatic than others.” In this context, considering that for Parkes, the grieving process involves assigning meaning to the loss, it becomes clear how the numerous pandemic disruptions (such as the conditions of the loss, the inability to say goodbye, diminished or absent funeral rituals, lack of social support, and especially the fear of one’s own contagion and death, as everyone was in a vulnerable situation) complicated this process of coming to terms with the loss and may have contributed to the overload on family members [39].

In the results of the present study, which conducted a more extensive and in-depth investigation into the meaning-making process of loss, it was observed that the unusual circumstances of the pandemic led to significant changes in the perception of time and experiences of space in such a way that the available spaces for grieving were confined to the private sphere, while time was perceived as extremely accelerated. In this sense, it was found that the bereaved made extreme efforts to try to make sense of the chaotic reality imposed by the pandemic. After all, grieving the loss of family members was experienced alongside other losses (financial, routine, health, security, autonomy, and future prospects).

Another important point was the absence or reduction in funeral rituals, especially wakes, and the opportunity to pay a final tribute to the deceased through the choice of clothing, the flowers to adorn the casket, and the presence of those coming to say goodbye [28,48,49]. In Brazilian culture, these rituals are very important as they serve three functions: saying farewell to the deceased, receiving community support, and, for many, participating in a religious rite of passage, tied to the hope of a more peaceful afterlife [28]. On the other hand, the participants’ testimonies highlight what Parkes [33] (p. 201) had already pointed out, “It is not enough […] to recommend a ritual. You have to believe in it”. Thus, rituals, which, according to the author, could offer some explanation and meaning for the death and provide social support for the expression of grief, were mechanically performed, not allowing people to assign meaning to their loss. As described by Hamid and Jahangir [50] and Hanna et al. [51], the impossibility of signifying farewell during a funeral and burial is apparent. Family members resented not only the distance from their loved ones during hospitalization but also not being able to see their bodies after death. At the most challenging moment of the pandemic, families received their deceased wrapped in plastic bags, not dressed, and with sealed coffins, factors perceived as a dehumanization of the deceased [45,46,50]. This raised questions about whose body was being mourned, leading to doubts about whether it was truly the family member in that coffin.

According to Parkes (2010) [33], it is expected that the bereaved will seek out the deceased relative, wait for their return, and continue with daily tasks as if they considered the presence of the loved one as part of the expected grieving process. In the scenario described above, the inability to personally confirm the death of the family member complicated the task of accepting the new reality, which took on surreal contours and could be used to convince oneself that this state of seeking the relative might contain elements of reality. Kentish-Barnes et al. [47] highlight the difficulty of family members getting in touch with their losses, defining the deprivation of spaces where one would authentically experience grief as stolen moments.

Another factor that contributed to the difficulty of the psychosocial transitions necessary for adaptation during the pandemic was the speed with which multiple events occurred, requiring many different changes in the daily lives of families. In this scenario, the exponential increase in deaths caused by COVID-19, which took place over a short period [9,10,11], impacted the participants. According to the literature, the pandemic impacted the perception of death as a sudden event, difficult to process [27,28,29,52,53,54,55]. However, the studies do not highlight the speed with which the events were experienced during the exceptional situation, either due to the rapid worsening of the clinical condition of the people affected or the obligation to perform burials urgently. Parkes (2010) [33] highlights that experiencing multiple deaths in the same family occurring within a brief period may hinder the process of working through losses. A lack of preparation for death, which could occur during the gradual period of anticipatory grief, can make it difficult to work through the loss during the grieving process. In the context, the sudden death often caused by COVID-19, in the pandemic context, seems to acquire more dramatic contours due to the unpredictability and accelerated pace of events [33].

The decreased accessibility to healthcare services (both in COVID-19 prevention and treatment), along with funeral services that were either canceled or shortened, represented significant additional stressors contributing to the emotional overload of the participants. They were unable to be physically present with their loved ones in hospitals or to participate in traditional mourning rituals. This perceived inaccessibility to care increased distress, as it deprived them not only of physical presence with the ill family member but also of the psychological comfort provided by familiar healthcare interactions and community mourning practices [56]. This deprivation of services, vital to both physical and emotional well-being, reflects broader social inequalities, as individuals from more vulnerable groups are generally disproportionately affected. Their limited ability to navigate or adapt to the altered care systems during the pandemic reveals how existing disparities were deepened, adding an additional layer of suffering and grief for many families.

Considering that more than half of the sample engaged in professional activities without formal employment ties, the need to both deny the necessity of protective isolation and to quickly reintegrate into the job market needs to be emphasized. In this way, psychological needs were reinforced by the need for subsistence. Alongside the numbness and denial inherent in the pandemic context were feelings of anguish and anger when recognizing that their family members were part of the sad statistic of anonymous victims. The participants related their discontent to political issues stemming from the poor management of the crisis, as the government’s negligence left them to fend for themselves, leaving them helpless in their experiences of loss while also being threatened by the out-of-control context exacerbated by denialist politics. The vaccines that could have prevented the deaths of their family members were not acquired deliberately [15,19]. The perception of the possibility of survival, given that the deaths occurred very close to the date when the deceased family members were scheduled to be vaccinated, intensified the suffering, indignation, and outrage of the survivors. For the research participants, the chance of survival was taken from their family members. If the government had managed the pandemic responsibly in the interest of the nation and in service of the Brazilian citizens, many deaths could have been avoided. If, nonetheless, the death of the family member had occurred, it could then be characterized as a tragedy rather than a crime conceived in the offices of central power occupied by the far right.

In this scenario of loneliness, discontent, and lack of support, the interviewees described how events continued to reverberate and be felt long after the death, reigniting moments of loss. As Parkes (2010) [33] noted, the painful memories of the final moments, which do not necessarily refer to the image of the deceased family member, are frequent due to the distance from the affected relatives. Thus, these memories are continuously revived as the participants hold on to painful and persistent recollections of the circumstances surrounding their losses, reporting episodes of crying, despair, longing, and a sense of meaninglessness in life. The violent and sudden nature of the pandemic, with massive losses, evokes a despair akin to that described in situations of natural disasters and major catastrophes, in which grief is interwoven with unexpected, terrifying, and violent events, characterized by multiple losses and the rupture of social systems that had previously provided support [33]. Therefore, the results obtained are consistent with Parkes (2006) [45] and also align with the literature, which indicates that grief experienced in disaster situations is permeated by intense feelings of anger, guilt, anxiety, discontent, shame, sadness, and shock.

It is noted, however, that despite all the suffering experienced, family members gradually manage to reach a degree of life organization and hope for the future, as proposed by Parkes when discussing the phase of adaptation and reorganization in grief, with the prospect of reopening social relationships and renewed hope brought by vaccination. This moment seems to be characterized as an opportunity to move out of stagnation, allowing individuals to oscillate between grief-related tasks (such as visiting the cemetery) and restoration tasks (such as social gatherings). Vaccination, in addition to the hope for the end of the pandemic, has brought the possibility of family reunions, which may be an important factor in the grieving process, making it no longer a solitary experience.

## 5. Conclusions

Although there is no specific term for “Brazilian grief,” it is important to consider the typical variables of this context to understand the phenomenon of grief: the denialism of the federal government, the exponential increase in the number of deaths each day, the healthcare system’s inability to care for all patients, the necessity of engaging in work activities while disregarding protective measures to ensure income for subsistence, the psychological impact resulting from the restriction of rituals that are extremely important in Brazilian culture, and the drastic reduction in social support. Parkes, in proposing the theory of Psychosocial Transitions, helps to understand the experiences during this period, but like any theoretical framework, it has limitations due to these and other cultural specificities.

Future research should focus on exploring cultural differences in grief, particularly how various cultures respond to loss during crisis situations and the effects of public health crises on mourning practices and rituals. Additionally, it is vital to assess the effectiveness of psychological interventions, such as group therapy and community support, in post-pandemic contexts, specifically examining how these interventions influence grief adaptation and emotional resilience. Moreover, analyzing the experiences of different demographic groups—such as age, socioeconomic status, and ethnicity—during health crises can uncover important nuances that can facilitate the development of tailored support interventions. Long-term studies examining the effects of grief initiated in the context of public health crises are essential for understanding how grief evolves over time, particularly regarding the persistence of grief symptoms and the coping strategies individuals employ.

The contributions of the present study pertain to the understanding of several potentially complicating factors related to grief and the difficulties in adapting to a new reality in a situation where a country (of continental dimensions) with no history of wars was experiencing, for the first time, the challenge of dealing with numerous deaths without governmental support. The findings of this study offer significant insights for grief interventions in future pandemics or other crisis contexts. First, they underscore the need to consider the cultural and social frameworks that shape grieving practices. In the Brazilian context, the denial of reality and the spread of misinformation had a profound impact on how individuals processed grief, highlighting the importance of timely and accurate public communication in crisis settings. For future interventions, ensuring clear communication and fostering social support networks, even if mediated by technology, can help individuals adapt more effectively to sudden losses.

Additionally, the study illustrates the importance of restoring mourning rituals, which play a crucial role in providing closure and support in certain cultures. The disruption of these rituals during the pandemic, such as limited access to funeral practices, intensified the suffering of bereaved individuals. Therefore, future interventions should prioritize maintaining or adapting cultural rituals, even in the face of restrictions, to offer individuals meaningful ways to honor their deceased loved ones. Psychosocial interventions that address both individual and collective needs, especially in communities disproportionately affected by public health crises, should also be developed. These interventions should consider the broader impact of grief on families, communities, and social roles, especially given the multifaceted nature of loss experienced during pandemics. Furthermore, the study emphasizes the importance of long-term grief support, as participants reported that feelings of loss and distress persisted long after the initial shock. This indicates that grief intervention programs should not be limited to the immediate aftermath of the crisis but should continue to offer support as individuals process their losses over time.

## Figures and Tables

**Table 1 ijerph-21-01398-t001:** Participants’ characterization (*n* = 10).

Characteristic, *n* (%)	Participants (*n* = 10)	Deceased Family Member (*n* = 16)
Sex		
Female	8 (80%)	7 (43.75%)
Male	2 (20%)	9 (56.25%)
Age (years)		
18–30	4 (40%)	1 (6.25%)
31–50	5 (50%)	3 (18.75%)
51–64	1 (10%)	4 (25%)
65+	0	8 (50%)
State of residence		
São Paulo	8 (80%)	15 (93.75%)
Minas Gerais	1 (10%)	1 (6.25%)
Bahia	1 (10%)	0
Occupation		
Formal employment	5 (50%)	0
Informal work	2 (20%)	3 (18.75%)
Small entrepreneur	0	3 (18.75%)
Student	1 (10%)	0
Unemployed	2 (20%)	1 (6.25%)
Retiree	0	9 (56.25%)
Relationship of deceased		
Grandmother	-	1 (6.25%)
Grandfather	-	2 (12.5%)
Mother	-	4 (25%)
Father	-	3 (18.75%)
Mother-in-law	-	1 (6.25%)
Father-in-law	-	1 (6.25%)
Brother	-	1 (6.25%)
Wife	-	1 (6.25%)
Husband	-	2 (12.5%)
Time since contagion and death (days)		
1–5	-	11 (68.75%)
6–10	-	4 (25%)
10+	-	1 (6.26%)
Funeral allowed		
Yes	-	6 (37.5%)
No	-	10 (62.5%)

**Table 2 ijerph-21-01398-t002:** Outline of themes and sub-themes.

Themes	Sub-Themes
Section 3.1. Living the anticipation of loss in an unknown world	Section 3.1.1. Transformations in Family Daily Life
	Section 3.1.2. Changes in Healthcare Settings
Section 3.2. Living through grief in a changed world	Section 3.2.1. Sudden Losses: reality took on unreal characteristics
	Section 3.2.2. Changes in Rituals and Accelerated Farewells
Section 3.3. Glimpsing a new possibility of living	Section 3.3.1. Vaccine: anger at the delay and hope for the vaccination itself
	Section 3.3.2. Glimpsing a new possibility of living

**Table 3 ijerph-21-01398-t003:** Qualitative evidence of the sub-theme Living the anticipation of loss in an unknown world.

Sub-Theme	Qualitative Evidence
Section 3.1.1. Transformations in Family Daily Life	I suffered in isolation. I was isolated in a bedroom in the back of the house. My mother wanted to stay close, so she’d appear at the window, and I’d say the whole time: ‘Mom, don’t come here’. (Roberta)
	The pandemic disunited families, especially in the moment of pain. If it weren’t for the pandemic, family members outside the city would certainly come, spend a few days with us, and comfort us. (Costa)
	When anyone in the family dies, we spend the weekend together. We try to deal with it together. So, I felt like we were very alone [in the pandemic]. (Regina)
	The worst feeling was that, even among us sisters and brothers, we couldn’t hug each other. We just had had a hell of a loss; we were still in shock with that situation, and we couldn’t comfort each other. (Antônia)
	They put a towel on her, like this [shows her bust], for me to hug her, they used alcohol. After I hugged her, they gave me alcohol and sprayed her with alcohol again. […] We were all in a state of shock. (Liz)
Section 3.1.2. Changes in Healthcare Settings	Not being able to help was nerve-racking, not being able to hear his voice, to see him. Not being able to touch him, being there. I’d think:] ‘Wow, what can I do to help?’ But there wasn’t much I could do. (Rita)
	I rushed to the hospital, so I caught him [husband] getting into the ambulance. He reached out and took my hand. (Reisa)
	I couldn’t stay long, I had to leave soon… I couldn’t hug her [mother] one more time. That’s what hurts me to this day. (Larissa)
	We got married when she was 16 years old. She died at 46, so I’ll say, it was her whole life by my side, and in the time of illness and death, I couldn’t be with her in the hospital. (Costa)
	At first, it was an online bulletin; it was very bad, very bad. Because we couldn’t see, we couldn’t do anything. I, personally, didn’t see my grandpa. (Regina)

**Table 4 ijerph-21-01398-t004:** Qualitative evidence of the sub-theme Living through grief in a changed world.

Sub-Theme	Qualitative Evidence
Section 3.2.1. Sudden losses: reality took on unreal characteristics	Every week we went to the cemetery, and there would be another ten burial sites. I remember being scared. It’s very complicated that he [grandfather] died of COVID-19 just because the world was dying of COVID-19, and I couldn’t deal with it. (Regina)
	Everything happened too quickly. The newspaper talked about a guy who was intubated for a month. Then, he left and is okay now. I said: ‘Wow, my mother stayed for two days and died.’ It’s a strange thing. (Larissa)
	I always say although the pain is enormous, we sometimes may understand a heart attack better, we may even understand cancer. But I still can’t understand this COVID-19. So, I think it’s a much greater burden. (Jessica)
	In the meantime, my mother-in-law died, my father-in-law, who was well, was hospitalized and then died. He [husband] didn’t know [about the death] of his mother or father. He spent 17 days in the ICU and then died. (Reissa)
	My wife died in a very short time. It all happened in a flash. Within two weeks, I was widowed. It was very fast, very fast. […] It was like snapping fingers. At one point, she [wife] was fine, but suddenly, she wasn’t. (Costa)
Section 3.2.2. Changes in rituals and accelerated farewells	It seems like it was just, really, leaving my grandfather there, and that was it. […] His body was inside a bag. So, everything was very complicated. We couldn’t see him at all. Everything was sealed. (Regina)
	There was no wake, it was just the burial. It was painful. My aunt still says to this day: ‘For me, [mother] is alive because I didn’t see her in the coffin. (Larissa)
	If I didn’t see his body, I don’t think I would ever accept it. Never. Never. I would think: ‘What if he’s alive. (Rebecca)
	We expect him to return from the grocery store at any time. That he will say it was a lie, a mistake. Because we didn’t see his body, we didn’t see him dead. (Liz)
	The pandemic makes it difficult for you to say goodbye to a relative who passes away” since “not being able to say goodbye, not being able to see. (Costa)

**Table 5 ijerph-21-01398-t005:** Qualitative evidence of the sub-theme Glimpsing a new possibility of living.

Sub-Theme	Qualitative Evidence
Section 3.3.1. Vaccine: anger at the delay and hope for the vaccination itself	COVID-19 was preventable. That’s what hurts me the most. We are in total chaos. My mother died because the president didn’t buy the vaccine that already existed. She died two weeks before she’d be vaccinated. (Romeo)
	My mother died from a disease for which there is already a vaccine. I am disgusted with this country and its leaders. Getting the vaccine that she [my mother] wanted to take connects me to her. So it gives me hope. (Larissa)
	I am feeling a mix of loneliness, sadness, and anger that they [father and grandmother] did not have the opportunity to get vaccinated. They could have been vaccinated today, but he didn’t make it. (Rita)
	It took a long time for the vaccine to come to us. If she had had the opportunity to get vaccinated, who knows, perhaps she would be alive. But this is ‘if’, ‘if’, ‘if’, ‘if’… So, now there’s no use. (Costa)
	People asked me, ‘Are you emotional? Are you happy that you took the vaccine?” and I said ‘I’m not. I’m not. I’m just relieved. I’m relieved, but at the same time, it’s unfortunate that he didn’t have time to take it. (Rebeca)
Section 3.3.2. Catching a glimpse of a new way of living	I’m in the dark; I don’t see a way out, I don’t see a door, I see nothing. I don’t know how to act, what to do. I know nothing. I don’t know where to start. What am I going to do? It’s tough. (Reisa)
	I don’t know where I’m going, I have no prospects. I suddenly found myself without my mother, and it seems that nothing I do from now on has meaning without her. (Jessica)
	I experienced a little freedom, to see and hug, and to let some tears flow in the presence of [his family members]. (Costa)
	Many things are changing in terms of perspectives of what is to come. I have many plans. (Regina)
	Little by little, I am getting back to life, starting to plan trips to meet loved ones and seek new opportunities to deal with my pain. (Rita)

## Data Availability

The data presented in this study are available upon request from the corresponding author. The data are not publicly available due to privacy or ethical restrictions.

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
