# Peer review of "Emotional Suffering After the COVID-19 Pandemic: Grieving the Loss of Family Members in Brazil"

_ijerph, 2024, doi:10.3390/ijerph21111398_

Round 1

Reviewer 1 Report

Comments and Suggestions for Authors

Comments on the Quality of English Language

it's ok

Author Response

R1.1.: Minor editing of English language required.

Thank you for your valuable corrections to the article and for the detailed comments on the manuscript. Your insights are greatly appreciated and will help improve the quality of the work. The entire text has been reviewed by a professional translator.

R1.2: It is an emotionally touching manuscript, which pertains to a very relevant bereavement topic associated with the COVID-19 pandemic in Brazil. The present study offers rich qualitative insights; however, amendments could be made with respect to academic rigour, coherence and presentation. Detailed comments follow whereby the strengths, weaknesses and areas of improvement are outlined.

We appreciate the recognition of the merit of the article and the valuable contributions provided, which will certainly enhance the quality of the work

R1.3: The manuscript discusses an under investigated issue: the grieving process in Brazil during the COVID-19 pandemic. While relevant, it is not duly framed in terms of how it contributes to the existing knowledge on grief and what unique ways the Brazilian context informs observations different from pandemic bereavement studies globally. It is expected that the distinctive contributions of the paper be articulated more clearly in the context of the literature on grief and bereavement. What is particular in the grieving in Brazil during COVID-19, which places Graves' study apart from all the work carried out so far? How does this research add to the knowledge of grief in extreme contexts?

We agree on the importance of better specifying the particularities of the COVID-19 pandemic in Brazil and how it impacts the grieving process. We have reviewed the text, taking your feedback into consideration. The unique context of the pandemic in Brazil, including high mortality rates, overwhelmed healthcare systems, and social inequalities, presents a distinct backdrop for mourning that differs from global experiences. Justifying this research within this specific scenario is crucial, as it highlights both the originality and relevance of the study. By doing so, we can more clearly demonstrate how this work adds to the existing literature on grief, particularly in extreme contexts, and contributes valuable insights into understanding bereavement in Brazil during such a critical period. We have added information on this topic to the manuscript, further clarifying the specific context of the COVID-19 pandemic in Brazil and its impact on the grieving process.

We have added the following information to the Introduction:

Thus, grief in Brazil during the pandemic was deeply intertwined with political and social issues that intensified the suffering of the bereaved. Political polarization divided the country and created rifts within families, making it difficult to provide the necessary social support during the mourning process. Additionally, inadequate access to healthcare systems and the economic crisis further deepened inequalities, heightening the sense of vulnerability.

[…]

Given these particularities, this article, by exploring grief in Brazil during the COVID-19 pandemic, highlights how extreme conditions—high mortality, misinformation, the collapse of the healthcare system, economic crisis, and social divisions—shaped a unique grieving experience with challenges and complexities distinct from those in other parts of the world. This study has the potential to contribute to the literature on grief in extreme crisis settings and social inequality.

We also revised the data discussion, aiming to better explore these social and political particularities

R1.4: The methodology is well described: the adopted approach is clinical-qualitative, descriptive exploratory. The preceding approach, however, has certain elements that can be fine-tuned. First, the nature of the sampling technique applied-snowball sampling-ought to be acknowledged ab initio as biased. Second, while the sample size is limited to 10 participants, justified by theoretical saturation, there is little reflection on ways in which the sample may limit generalization of the study's findings. Snowball sampling has its limitations, and limitations regarding small sample size are foreseen as reducing generalizability. The authors also need to provide a discussion of other possible sampling methods that might have enhanced the representativeness of the sample.

Thank you for pointing out the limitations of the snowball sampling method. We have added a paragraph acknowledging the potential biases inherent in the use of this technique and emphasizing the issues related to the generalizability of the results. Our focus is on obtaining in-depth qualitative insights from the specific experiences of the participants. We appreciate the feedback and will clarify these points in the manuscript to enhance the methodology.

Added to the text:

The snowball sampling method is widely used in qualitative research, where the researcher's focus is not on numerical representativity, and there is no concern with random selection. The goal is to recruit participants who can offer privileged information on the topic. The focus is not on controlling variables to increase the generalizability of the results, but rather on gaining an in-depth understanding of a phenomenon from the perspective of the participants. One advantage of this method is that it fosters an environment of trust, as the participants are contacted through the recommendation of someone they know. On the other hand, one disadvantage is the risk of limited diversity in the sample, as participants often share similar characteristics (a phenomenon known as ‘anchoring’). To mitigate these potential biases, certain precautions were taken in selecting participants, following guidelines from the literature (Kirchherr & Charles, 2018), such as ensuring diversity among the initial participants, not limiting data collection to a single dissemination of the study, and promoting the research through various channels and contexts.

Reference

Kirchherr J, Charles K. Enhancing the sample diversity of snowball samples: Recommendations from a research project on anti-dam movements in Southeast Asia. PLoS One. 2018 Aug 22;13(8):e0201710. doi: 10.1371/journal.pone.0201710.

R1.5: The authors used Parkes' theory of mourning as the theoretical framework, and it is very applicable for this topic. However, the integration of theory with empirical findings is rather superficial. That is, not enough deep discussion on how findings relate or challenge the theory can be found. Please integrate the theoretical framework deeper with the findings. Rather than referencing Parkes' theory, critically engage with the theory of Parkes; you might even provide some insight about how the Brazilian context or special conditions of the pandemic challenge or support his theory.

Thank you for your insightful comments regarding the integration of Parkes' theory of mourning with our empirical findings. We appreciate your suggestion to critically engage with the theory rather than simply referencing it. In response, we have reanalyzed our data, taking your feedback into consideration. The revised manuscript now includes a deeper integration of the theoretical framework with our findings, as well as insights into how the Brazilian context and the unique conditions of the pandemic both challenge and support Parkes' theory.

R1.6. The data collection procedure is highlighted through semi-structured interviews. The application of thematic analysis is proper for this qualitative study. However, the manuscript does not highlight how further data processing has been done apart from thematic analysis. For example, how were themes identified? Was there coding? This area of the analysis can be enhanced by making clear how the themes came up from the data.

Thank you for your valuable feedback regarding the data collection and analysis procedures. We have added more information in the data analysis section of the manuscript to clarify how themes were identified and the coding process used. This additional detail aims to enhance the transparency of our thematic analysis and provide a clearer understanding of how the themes were created from the data. Added to text:

The researchers imported the interview material into the QDA software, which provided a variety of tools for editing and grouping the analyses. After conducting thorough readings, the researchers were able to create and organize the codes, exporting excerpts into spreadsheets that formed the categories and/or subcategories.

R1.7. The results are well presented with poignant quotes from participants, but the presentation of results could be more structured, like a themed discussion. For instance, some themes, such as "Death in altered places: Physical isolation and emotional loneliness," could relate further to the literature. Also, quotes, while evocative, at times feel overwhelming in relation to the analysis. Smarter balance between quotation from participants and the analysis; quotes are used to illustrate points, not to lead the narrative. Discussion of each of the themes requires further development through critical use of the literature.

Thank you for your thoughtful comments on the presentation of the results. We appreciate your suggestions for structuring the results into a more thematic discussion. We've grouped the categories together, trying to give the text more fluidity and cohesion, and we've shortened the participants' speeches, which are now primarily intended to illustrate the results. In addition, the discussion was reworked, seeking greater dialog with studies in the area and with Parkes' theory.

R1.8. The discussion synthesizes the findings well; however, it needs to be more critical. There is room for discussion of the wider implications of these findings, including a relation to the prospects for mental health support within the pandemic or crisis contexts in Brazil and globally. The authors also need to delve more deeply into the political context alluded to by participants, which might have set a context within which the grieving process was influenced by governmental responses.

Thank you for your insightful feedback on the discussion section. We appreciate your suggestion to adopt a more critical perspective and to explore the wider implications of our findings, particularly regarding mental health support in pandemic or crisis contexts both in Brazil and globally. We have incorporated these suggestions into the revised manuscript, delving deeper into the political context mentioned by participants and examining how governmental responses may have influenced the grieving process. Your input has been invaluable in enhancing the depth and relevance of our discussion

R1.9. While the authors do point out some limitations, like small sample size and one geographic location, this section could be a bit more thorough. For example, no statement is made regarding sample homogeneity, which could include socioeconomic status, race, or religion.

Thank you for your constructive feedback regarding the limitations section. We appreciate your suggestions for making this section more thorough. In response, we have rewritten the limitations to include considerations regarding the method used to recruit participants.

A more concise sample enabled a more in-depth and detailed analysis of the experiences of loss in the context of the pandemic. However, there are some limitations since most participants lived in a single Brazilian state, and the interviews were conducted in a particular context. This method may contribute to a certain level of homogeneity and limited variability among respondents. These data may be related to the participant recruitment procedure, a topic already addressed in the method, which may contribute to a homogenization and low variability among respondents, even though methodological precautions were taken to reduce these impacts.

R1.10. The conclusion, even though well-written, lacks a look into the future. The authors do summarize the findings but fail to indicate recommendations that might be addressed by professionals in mental health, policy makers, and even researchers. Additionally, potential further research is underdiscussed.

R: Thank you for your valuable feedback on the conclusion section. We appreciate your insights regarding the need for a forward-looking perspective. In response, we have rewritten the conclusions to include recommendations for mental health professionals, policymakers, and researchers. Additionally, we have expanded the discussion on potential avenues for further research. The following information has been added:

Although there is no specific term for "Brazilian grief," it is important to consider the typical variables of this context to understand the phenomenon of grief: the denialism of the federal government, the exponential increase in the number of deaths each day, the healthcare system's inability to care for all patients, the necessity of engaging in work activities while disregarding protective measures to ensure income for subsistence, the psychological impact resulting from the restriction of rituals that are extremely important in Brazilian culture, and the drastic reduction in social support. Parkes, in proposing the theory of Psychosocial Transitions, helps to understand the experiences during this period, but like any theoretical framework, it has limitations due to these and other cultural specificities.

The contributions of the present study pertain to the understanding of several potentially complicating factors of grief and the difficulties of adapting to the new reality in a situation where a country of continental dimensions and without a history of wars was facing, for the first time, the challenge of dealing with numerous deaths without governmental support. It was found that grief, as a longitudinal process, continues to be experienced and psychologically processed long after a loss. Therefore, to support interventions in the crisis context and beyond, new studies are needed to consider the specificities and late effects of grief initiated in the pandemic context. It is also suggested that studies compare the realities experienced in different countries, considering the influence of social, cultural, and economic factors on the experience of grief.

R1.11. The manuscript is generally well-written; some parts are a bit dense and need streamlining to ensure clarity. There are also occasional grammatical errors and awkward phrasing which will need attention in a thorough language edit.

R: Thank you for your feedback regarding the clarity and overall quality of the manuscript. We appreciate your observations about the density of certain sections and the grammatical errors. In response, the text has undergone a new review by a professional editor to address these issues and ensure improved clarity and readability. Your input has been instrumental in enhancing the quality of our work.

Reviewer 2 Report

Comments and Suggestions for Authors

Dear authors, I really appreciated your manuscript for the care and effort, it is well structured, complete in almost all parts, adequately written and with the correct references, but there are some critical issues that weaken the study design and discussions, and deserve to be revised:

1) the entire manuscript lacks statistical analysis of the results, which would give strength to the discussions;

2) the graphical parts of tables and graphs are missing, depending on the results and discussions;

3) the discussions and conclusions do not take into account the multifactorial nature of the investigation you have completed, as they are only reported in discussions and conclusions but without a backing of detailed results. I suggest that you go into these sections in detail, appropriately including all the variables involved, identifying them individually and in their comparisons. 

Author Response

Revisor 2

Dear authors, I really appreciated your manuscript for the care and effort, it is well structured, complete in almost all parts, adequately written and with the correct references, but there are some critical issues that weaken the study design and discussions, and deserve to be revised:

(x) English language fine. No issues detected.

1) the entire manuscript lacks statistical analysis of the results, which would give strength to the discussions;

2) the graphical parts of tables and graphs are missing, depending on the results and discussions;

3) the discussions and conclusions do not take into account the multifactorial nature of the investigation you have completed, as they are only reported in discussions and conclusions but without a backing of detailed results. I suggest that you go into these sections in detail, appropriately including all the variables involved, identifying them individually and in their comparisons.

Thank you very much for your kind words and feedback. We have added a specific section explaining the absence of statistical analysis. Additionally, a table has been included in the participants and results section, and the discussion of the results has been expanded to cover a broader range of variables.

No formal statistical hypothesis testing or sample size calculation was performed; all reported data are qualitative and descriptive and categorical variables were reported as counts (frequencies and percentages) The sample size of 10 family members was considered reasonable to allow for the identification of key themes in the results. As a clinical-qualitative study, there was no intention to generalize the findings.

Reviewer 3 Report

Comments and Suggestions for Authors  

Certainly! Here’s a revised version of your text for clarity and grammar:

Dear Authors,

I would like to extend my congratulations on your work, which I found very compelling. As a researcher and clinical psychologist working in this field, I want to emphasize the importance of addressing emotional suffering due to loss, particularly during global emergencies. Some of your conclusions resonate with a study I authored some time ago, which explores the experience of loss in the context of the disruption to normalcy brought by the pandemic (https://link.springer.com/article/10.1007/s11089-021-00991-0).

Returning to your paper, I would recommend mentioning the four themes in the abstract (it would be sufficient to simply name them). This would allow researchers to gain a clearer perspective on your findings at a glance.

Regarding the introduction, when introducing Parkes’ perspective, I suggest incorporating some significant results from other qualitative studies conducted in similar contexts.

Additionally, I would recommend rephrasing the final paragraph, as it is somewhat difficult to read.

The material and methods section is well written. However, I do not understand why interviews were also video-recorded (as it not seems, in this section, that you value that type of data). Moreover, there is the need to provide some examples of your guiding questions. On another note, I think there is the need to say more about the theoretical framework and why/how you choose it. There are many interesting theories and perspective in the field, and it would be appropriate to share more about your decision. Finally, I would create an appropriate participants section in the material and methos macro-section where I would add the information provided at the beginning of the Results section, as they seem more pertinet to material and methods.

Apart from that, I then think that results are clearily presented and the discussion is well-informed.

Comments on the Quality of English Language

I would suggest to re-phrase the final section of the Introduction as it seems more difficult.

Author Response

Revisor 3

R3: Minor editing of English language required.

Thank you for your valuable corrections to the article. Your insights are greatly appreciated and will help improve the quality of the work. The entire text has been reviewed by a professional translator.

R3.2. I would like to extend my congratulations on your work, which I found very compelling. As a researcher and clinical psychologist working in this field, I want to emphasize the importance of addressing emotional suffering due to loss, particularly during global emergencies. Some of your conclusions resonate with a study I authored some time ago, which explores the experience of loss in the context of the disruption to normalcy brought by the pandemic (https://link.springer.com/article/10.1007/s11089-021-00991-0).

Thank you for your thoughtful comment and for recommending the reading. I truly appreciate your insights, especially as they relate to addressing emotional suffering due to loss during global emergencies. We are currently working on an article that examines the role of spiritually in the grieving process, and your study will certainly be a valuable addition to our research.

R3.3. Returning to your paper, I would recommend mentioning the themes in the abstract (it would be sufficient to simply name them). This would allow researchers to gain a clearer perspective on your findings at a glance.

Thank you for your valuable suggestion regarding the abstract. We appreciate your recommendation to mention the four themes, as it provides researchers with a clearer perspective on our findings. In response, we have revised the abstract to include the names of the themes, enhancing its clarity and comprehensiveness.

Results: The results were organized into two categories: Death in altered places: Physical isolation and emotional loneliness and Death in an accelerated time: aborted farewells.

R3.4. Regarding the introduction, when introducing Parkes’ perspective, I suggest incorporating some significant results from other qualitative studies conducted in similar contexts.

Thank you for your suggestion regarding the introduction. We chose to incorporate Parkes' perspective, as it serves as a theoretical framework for discussing our data. However, we appreciate your recommendation to include results from other qualitative studies conducted in similar contexts. In response, we have added data from a metasynthesis carried out by Brazilian researchers on the topic. 

A meta-synthesis conducted with the objective to summarize and reinterpret the results of qualitative studies on the experience of losing family members during the pandemic by a thematic synthesis. It indicated indicate that experiences of loss in this context were negatively impacted by the imperatives of physical distance, restriction of hospital visits, technology-mediated communication, and prohibition or restriction of funerals. These changes resulted in experiences marked by feelings of loneliness and helplessness, which should be considered when planning intervention strategies that favor communication between family members with the afflicted loved one and with the health care team, enabling welcoming and creating alternatives for farewell rituals [29].

R3.5. Additionally, I would recommend rephrasing the final paragraph, as it is somewhat difficult to read.

Thank you for your feedback regarding the final paragraph. We have reformulated it to enhance clarity and improve readability, ensuring that the information is presented more effectively.

Given these particularities, this article, by exploring grief in Brazil during the COVID-19 pandemic, highlights how extreme conditions—high mortality, misinformation, the collapse of the healthcare system, economic crisis, and social divisions—shaped a unique grieving experience with challenges and complexities distinct from those in other parts of the world. This study has the potential to contribute to the literature on grief in extreme crisis settings and social inequality. Understanding the problems caused by the pandemic in the grieving processes, considering the prolonged crisis, might contribute to the care provided to bereaved people and design strategies to cope with future pandemics by considering the needs of grieving populations. Bereavement persists long after death; thus, research remains relevant and necessary, especially in the Brazilian context, where there is a scarcity of qualitative research published in widely circulated journals [29]. For this reason, this study aims to understand the emotional suffering after the COVID-19 pandemic in Brazilian adults whose family members have died due to COVID-19.

R3.6. The material and methods section is well written. However, I do not understand why interviews were also video-recorded (as it not seems, in this section, that you value that type of data).

Thank you for your question regarding the video recordings of the interviews. The interviews were conducted online, and the platform we used only allowed for video recording. However, we utilized only the audio from these recordings for our analysis. We appreciate your attention to this detail, and we will clarify this point in the manuscript to avoid any confusion.

R3.7. Moreover, there is the need to provide some examples of your guiding questions.

Thank you for your suggestion regarding the inclusion of guiding questions. We have added examples of the guiding questions to the manuscript to provide clarity and enhance understanding of our interview process.

Examples of questions include: Can you share about your relationship with the family member before their illness?   How was the diagnosis communicated? What significant experiences did you have during the treatment? How was the news of the death conveyed? Was there a burial? How are you feeling at the moment? Do you have any plans for the future?

R3.8. On another note, I think there is the need to say more about the theoretical framework and why/how you choose it. There are many interesting theories and perspective in the field, and it would be appropriate to share more about your decision. 

Thank you for your valuable feedback regarding the theoretical framework. We have added more information to the manuscript explaining our rationale for choosing this particular framework and how it aligns with our study.

The theoretical framework of this study is based on the theory of grief proposed by Parkes (2010) [33]. According to this author, everyone constructs a worldview that involves habits and thoughts, which need to be revised after significant changes, a process he calls Psychosocial Transition. Losses create discrepancies between the internal world and the world that comes into existence, turning a once familiar world into something suddenly unknown. According to Parkes, some events are so impactful that they can invalidate entire areas of our assumed world, leading individuals to experience the feeling that nothing is certain anymore, making it very difficult to process these significant changes. The events that pose the greatest challenge for adaptation are those that require a reassessment of one’s worldview, involve lasting (rather than temporary) changes, and occur within a short period of time. In this context of multiple losses, as was the case during the COVID-19 pandemic, unexpected deaths combined with various other forms of grief can undermine the assumption that the world is a safe place. Changes face resistance, as individuals cling to their old model of the world, and in many cases, support networks and protective measures are necessary to aid the individual during this transition process.

R.3.9. Finally, I would create an appropriate participants section in the material and methods macro-section where I would add the information provided at the beginning of the Results section, as they seem more pertinet to material and methods.

Thank you for your comment, and we have accepted the suggestion. The description of the participants now appears in the methods section, rather than in the results

  1. Apart from that, I then think that results are clearily presented and the discussion is well-informed.

We appreciate your positive feedback on the clarity of the results and the quality of the discussion. Your encouragement means a lot to us as we strive to enhance our work.

Round 2

Reviewer 1 Report

Comments and Suggestions for Authors

Some of the responses are not in English. I cannot read it. So I have to read it through again. Please do not do that again.

1.        The background provides a comprehensive view of the COVID-19 pandemic's impact in Brazil, but it could benefit from more comparative context (e.g., Stroebe & Schut, 2021). How does the situation in Brazil differ from other countries, especially regarding grief and mourning practices? Adding this would enhance the global relevance of the paper.

2.        The introduction tends to overemphasize Brazil's political situation and government response to the pandemic. While this is important, the discussion about grief should be more balanced between the political, social, and psychological aspects of the experience.

3.        The rationale for selecting grief theory, specifically Parkes’ theory of mourning, is clear, but the manuscript could explore additional frameworks or compare multiple theories to provide a richer understanding of the grief process (Kauffman, 2013; Neimeyer, 2001).

4.        The sample size of ten participants is quite small. While qualitative studies often work with smaller samples, the generalizability of the findings might be questioned. It would be helpful to justify the sample size more explicitly by comparing it to similar studies.

5.        There is little discussion on the ethical challenges of conducting online interviews about such sensitive topics. How did the researchers ensure the emotional safety of participants, especially given the heavy and personal nature of the discussions?

6.        The results are well-organized into two major themes, but there is some repetition in the sub-themes. For instance, the discussion of physical isolation and emotional loneliness overlaps with the accelerated time of farewells. Consider restructuring to avoid redundancy.

7.        The presentation of results heavily relies on personal anecdotes, which, while powerful, may overshadow a broader analysis. It would strengthen the paper to intersperse these quotes with more analytical commentary to tie them back to the literature.

8.        The discussion is thorough, but there is a risk of being too descriptive. While it addresses the findings effectively, the manuscript could engage more critically with the implications of the research. What do these findings mean for grief interventions in other contexts or future pandemics?

9.        One important aspect to consider is how the theme of death in altered places and the issues of emotional loneliness and restricted access during the pandemic tie into perceived accessibility of essential public services, such as healthcare and funeral services. The pandemic drastically affected the accessibility of these services due to lockdowns, overwhelmed health systems, and strict safety protocols. This not only limited the ability of families to visit their loved ones in hospitals or attend funerals but also contributed to a heightened sense of isolation and helplessness.

From a perceived accessibility perspective (Liu et al., 2022), many participants likely felt that access to healthcare and the ability to engage in traditional mourning practices were diminished. The inaccessibility of these services might have worsened their emotional distress, as they were not only deprived of direct physical presence with loved ones but also of the psychological support provided by normal healthcare interactions and community-based mourning rituals. This disruption in perceived accessibility of crucial public services underscores the broader social inequities exposed by the pandemic, as it disproportionately affected those who could not easily adapt to or navigate the altered systems of care and support.

10.     The authors might consider expanding their reflection on the limitations of Parkes’ grief theory. For example, how does it account for collective grief or societal trauma in a pandemic, and are there alternative theories that could complement or challenge Parkes’ perspective?

11.     The conclusion could benefit from a stronger closing statement on the potential for future research. What specific areas should be explored next, particularly in relation to grief and public health crises?

12.     There are minor inconsistencies in referencing styles. Ensure all citations are formatted uniformly according to journal guidelines.

Reference

Kauffman, J. (Ed.). (2013). Loss of the assumptive world: A theory of traumatic loss. routledge.

Liu, Q., Liu, Z., Lin, S., & Zhao, P. (2022). Perceived accessibility and mental health consequences of COVID-19 containment policies. Journal of Transport & Health25, 101354.

Neimeyer, R. A. (2001). Meaning reconstruction & the experience of loss. American Psychological Association.

Stroebe, M., & Schut, H. (2021). Bereavement in times of COVID-19: A review and theoretical framework. OMEGA-Journal of Death and Dying82(3), 500-522.

Comments on the Quality of English Language

its ok

Author Response

Comments 1: The background provides a comprehensive view of the COVID-19 pandemic's impact in Brazil, but it could benefit from more comparative context (e.g., Stroebe & Schut, 2021). How does the situation in Brazil differ from other countries, especially regarding grief and mourning practices? Adding this would enhance the global relevance of the paper.

Response 1: In the first revision of the article, we included political and economic aspects of the Brazilian context that may influence the experience of grief. In this second revision, we added information from the article “The effect of suppressing funeral rituals during the COVID-19 pandemic on bereaved families,” developed in the national context and referenced in the article cited by the reviewer. The following information was added to the introduction (page 2).

In the Brazilian context, paying last respects to a loved one is a deeply cultural act, not only providing an opportunity to honor and say farewell to the deceased but also to receive support from the community. When these rituals cannot be performed, mourners often report feeling that the grieving process is left incomplete, and the inability to share their pain intensifies their sense of loss [28].

Comments 2: The introduction tends to overemphasize Brazil’s political situation and government response to the pandemic. While this is important, the discussion about grief should be more balanced between the political, social, and psychological aspects of the experience.

Response 2: Additional political aspects were introduced in the introduction in response to a reviewer’s suggestion. We chose to reduce what seemed like an overemphasis on political issues, while still highlighting the significant role we believe government management played during the pandemic. The following paragraph was removed:

Thus, grief in Brazil during the pandemic was deeply intertwined with political and social issues, which exacerbated the suffering of the bereaved. Political polarization divided the country and caused rifts within families, making it difficult to provide the necessary social support during the mourning process. Additionally, inadequate access to healthcare systems and the economic crisis further deepened inequalities, heightening the sense of vulnerability.

Comments 3: The rationale for selecting grief theory, specifically Parkes’ theory of mourning, is clear, but the manuscript could explore additional frameworks or compare multiple theories to provide a richer understanding of the grief process (Kauffman, 2013; Neimeyer, 2001).

Response 3: To select the theoretical framework on grief, we conducted a narrative review with the aim of synthesizing the main theoretical systems that support the interventions of health professionals. We highlighted the key concepts from the contributions of Freud, Kübler-Ross, Bowlby, Parkes, Worden, Neimeyer, Klass, Stroebe, and Schut, who are the most referenced authors in the Brazilian context. This review is currently under evaluation for publication in a scientific journal. We acknowledge that all the theories could assist in understanding the grief phenomenon, but we chose Parkes’ theory for two reasons: it is widely used by Brazilian researchers and he has a partnership with one of the leading national researchers on the subject (Prof. Dr. Maria Helena Pereira Franco). Additionally, we found that his theory aligns with our findings in previous works. Reflecting on the possibility of exploring additional aspects, we added the concept of overload to the introduction, which was introduced in 2016 in the dual process model. This concept refers to the grieving person's perception of having more than they feel capable of handling – whether too many activities, events, experiences, or other stimuli –because we believe it helps explain some of our findings.

As a complement to Parkes’ theory, we applied the concept of overload from the dual process model, recognizing that in a context of numerous changes and the need for readaptation, the grieving person may feel overwhelmed by more issues than they can manage. In this scenario, the coping process, which involves oscillating between loss-oriented and restoration-oriented stressors, may not take place. This is because the burden of one or both of these poles, or even additional stressors unrelated to the grief itself, can become too intense and overwhelming [39].

We add to the discussion:

It is noted, however, that despite all the suffering experienced, family members gradually manage to reach a degree of life organization and hope for the future, as proposed by Parkes when discussing the phase of adaptation and reorganization in grief, with the prospect of reopening social relationships and renewed hope brought by vaccination. This moment seems to be characterized as an opportunity to move out of stagnation, allowing individuals to oscillate between grief-related tasks (such as visiting the cemetery) and restoration tasks (such as social gatherings). Vaccination, in addition to the hope for the end of the pandemic, has brought the possibility of family reunions, which may be an important factor in the grieving process, making it no longer a solitary experience.

Reference

Stroebe M., & Schut H. (2016). Overload: A missing link in the dual process model? OMEGA: Journal of Death and Dying, 74, 96–109. https://doi.org/10.1177/0030222816666540

Comments 4: The sample size of ten participants is quite small. While qualitative studies often work with smaller samples, the generalizability of the findings might be questioned. It would be helpful to justify the sample size more explicitly by comparing it to similar studies.

Response 4: We understand that, in qualitative research, the purpose is to recruit participants who can provide relevant information on the topic under investigation. The focus is not on controlling variables to enhance the generalizability of the results but rather on gaining an in-depth understanding of a phenomenon from the participants' perspectives. Based on a recent metasynthesis (Sola et al., 2023) that analyzed qualitative research methodologies, it was observed that, out of the 14 included articles, nine (64%) were conducted with samples of up to 20 participants. Of these, five studies (35%) recruited ten or fewer participants, with four studies including samples of fewer than seven family members. The average number of participants in the analyzed articles was 29, with samples ranging from two to 196 participants. Excluding the study with the largest number of participants, the average number of participants in the remaining 13 articles was 15.07. Therefore, the inclusion of ten participants in this research is not an exception compared to other published qualitative studies.

Reference (metasynthesis)

Sola, P.P.B., Souza, C., Rodrigues, E.C.G., Santos, M.A., & Oliveira-Cardoso, E.A. (2023). Family grief during the COVID-19 pandemic: A meta-synthesis of qualitative studies. Reports in Public Health, 39(2), 1-20. https://doi.org/10.1590/0102-311XEN058022.

References (articles with samples smaller than 10 participants - n < 10)

Chen C, Wittenberg E, Sullivan SS, Lorenz RA, Chang YP. The experiences of family members of ventilated COVID-19 patients in the intensive care unit: a qualitative study. Am J Hosp Palliat Care 2021; 38:869-76.

Cordero Jr DA. Sákit Pighati and Pag-asa: a pastoral reflection on suffering during the COVID-19 pandemic in the Philippines. J Relig Health 2021; 60:1521-42.

Hernández-Fernández C, Meneses-Falcón C. I can’t believe they are dead. Death and mourning in the absence of goodbyes during the COVID-19 pandemic. Health Soc Care Community 2023; 30:e1220-32.

Tay DL, Thompson C, Jones M, Gettens C, Cloyes KG, Reblin M, et al. "I feel all alone out here": analysis of audio diaries of bereaved hospice family caregivers during the COVID-19 pandemic. J Hosp Palliat Nurs 2021; 23:346-53.

Wong LP, Tan SL, Alias H, Sai TE, Saw A. Psychological consequences of the delay in the Silent Mentor Programme during the COVID-19 pandemic: perspectives from family members of silent mentors. OMEGA (Westport) 2021; [Online ahead of print].

Comments 5: There is little discussion on the ethical challenges of conducting online interviews about such sensitive topics. How did the researchers ensure the emotional safety of participants, especially given the heavy and personal nature of the discussions?

Response 5: We acknowledge that interviews with bereaved participants involve ethical considerations that must be addressed, particularly in the context of technology-mediated interviews. As mentioned in the manuscript, the option of a group psychological intervention was offered to the participants. The interviews took place in July 2021, and the group, coordinated by three psychology professionals, ran from August to December 2021, totaling 20 sessions, each lasting approximately 80 minutes. The results of this intervention were published in a journal (omitted to preserve the anonymity of the authors).

The following information was added:

The online interviews were conducted during a time when the Federal Council of Psychology had established guidelines for psychological care mediated by technology, and these instructions were used to guide the interviews. At the end of the sessions, participants were asked how they experienced the contact, and all affirmed that it was a moment in which they felt welcomed and had a safe space to express their pain. Furthermore, to uphold the principles of beneficence, non-maleficence, and justice, a free online therapeutic group was offered to participants whose need for continued care was identified [41,42]. Nine of the ten interviewees participated in the group, with only one family member declining the offer [41,42].

References

  1. Parkes, C.M. (1995). Guidelines for conducting ethical bereavement research. Death Studies, 19(2), 171-181. https://doi.org/10.1080/07481189508252723.
  2. Sola, P.P.B.,Garcia, J.T., Santos, J.H., Santos, M.A., & Oliveira-Cardoso, E.A. (2022). Grupo online para familiares enlutados durante a pandemia no contexto brasileiro. Psicologia, Saúde & Doenças, 23(2), 390-397. https://doi.org/10.15309/22psd230205

Comments 6: The results are well-organized into two major themes, but there is some repetition in the sub-themes. For instance, the discussion of physical isolation and emotional loneliness overlaps with the accelerated time of farewells. Consider restructuring to avoid redundancy.

Response 6: We agreed that the results were redundant in some categories; therefore, they were reorganized into three mutually exclusive and complementary categories.

Table 2. Outline of Themes and Sub-themes.

Themes

Sub-themes

1. Living the anticipation of loss in an unknown world

1.1 Transformations in Family Daily Life

1.2 Changes in Healthcare Settings

2. Living through grief in a changed world

2.1 Sudden Losses: reality took on unreal characteristics

2.2 Changes in Rituals and Accelerated Farewells

3. Glimpsing a new possibility of living

3.1. Vaccine: anger at the delay and hope for the vaccination itself

3.2. Glimpsing a new possibility of living

Comments 7: The presentation of results heavily relies on personal anecdotes, which, while powerful, may overshadow a broader analysis. It would strengthen the paper to intersperse these quotes with more analytical commentary to tie them back to the literature.

Response 7: Following the guidance of another reviewer, the longer statements were organized into a table format to make the reading of the results more fluid and less solely based on the participants' narratives.

Comments 8: The discussion is thorough, but there is a risk of being too descriptive. While it addresses the findings effectively, the manuscript could engage more critically with the implications of the research. What do these findings mean for grief interventions in other contexts or future pandemics?

Response 8: Thank you for recognizing the thoroughness and quality of the discussion. We appreciate your feedback and have now added suggestions for grief interventions based on the findings. These suggestions focus on developing culturally sensitive support strategies, ensuring access to mourning rituals, and improving communication during public health crises. Additionally, the implications for psychosocial care in future pandemics have been addressed, with emphasis on tailored interventions that consider the unique challenges of grief in crisis contexts.

The findings of this study offer significant insights for grief interventions in future pandemics or other crisis contexts. First, they underscore the need to consider the cultural and social frameworks that shape grieving practices. In the Brazilian context, the denial of reality and the spread of misinformation had a profound impact on how individuals processed grief, highlighting the importance of timely and accurate public communication in crisis settings. For future interventions, ensuring clear communication and fostering social support networks, even if mediated by technology, can help individuals adapt more effectively to sudden losses.

Additionally, the study illustrates the importance of restoring mourning rituals, which play a crucial role in providing closure and support in certain cultures. The dis-ruption of these rituals during the pandemic, such as limited access to funeral practices, intensified the suffering of bereaved individuals. Therefore, future interventions should prioritize maintaining or adapting cultural rituals, even in the face of restrictions, to offer individuals meaningful ways to honor their deceased loved ones. Psychosocial interven-tions that address both individual and collective needs, especially in communities dis-proportionately affected by public health crises, should also be developed. These inter-ventions should consider the broader impact of grief on families, communities, and social roles, especially given the multifaceted nature of loss experienced during pandemics. Furthermore, the study emphasizes the importance of long-term grief support, as par-ticipants reported that feelings of loss and distress persisted long after the initial shock. This indicates that grief intervention programs should not be limited to the immediate aftermath of the crisis but should continue to offer support as individuals process their losses over time.

Comments 9: One important aspect to consider is how the theme of death in altered places and the issues of emotional loneliness and restricted access during the pandemic tie into perceived accessibility of essential public services, such as healthcare and funeral services. The pandemic drastically affected the accessibility of these services due to lockdowns, overwhelmed health systems, and strict safety protocols. This not only limited the ability of families to visit their loved ones in hospitals or attend funerals but also contributed to a heightened sense of isolation and helplessness.

From a perceived accessibility perspective (Liu et al., 2022), many participants likely felt that access to healthcare and the ability to engage in traditional mourning practices were diminished. The inaccessibility of these services might have worsened their emotional distress, as they were not only deprived of direct physical presence with loved ones but also of the psychological support provided by normal healthcare interactions and community-based mourning rituals. This disruption in perceived accessibility of crucial public services underscores the broader social inequities exposed by the pandemic, as it disproportionately affected those who could not easily adapt to or navigate the altered systems of care and support.

Response 9: We appreciate the suggestion of the article and have included the following text, based on this reading:

           The decreased accessibility to healthcare services (both in COVID-19 prevention and treatment), along with funeral services that were either canceled or shortened, represented significant additional stressors contributing to the emotional overload of the participants. They were unable to be physically present with their loved ones in hospitals or to participate in traditional mourning rituals. This perceived inaccessibility to care increased distress, as it deprived them not only of physical presence with the ill family member but also of the psychological comfort provided by familiar healthcare interactions and community mourning practices [56]. This deprivation of services, vital to both physical and emotional well-being, reflects broader social inequalities, as individuals from more vulnerable groups are generally disproportionately affected. Their limited ability to navigate or adapt to the altered care systems during the pandemic reveals how existing disparities were deepened, adding an additional layer of suffering and grief for many families.

Comments 10: The authors might consider expanding their reflection on the limitations of Parkes’ grief theory. For example, how does it account for collective grief or societal trauma in a pandemic, and are there alternative theories that could complement or challenge Parkes’ perspective

Response 10: We appreciate the reviewer's thoughtful feedback and fully agree with the suggestions provided. We have expanded our reflection on the limitations of Parkes’ grief theory and have included recommendations for future studies that consider aspects of collective grief, ambiguous loss, or necropolitics.

Comments 11: The conclusion could benefit from a stronger closing statement on the potential for future research. What specific areas should be explored next, particularly in relation to grief and public health crises?

Response 11: We appreciate the suggestion to strengthen the conclusion with a more assertive statement about the potential for future research. We recognize that the experiences of grief during the COVID-19 pandemic presented unique complexities that require ongoing investigation. To enrich this aspect, we suggest the following specific areas for future research:

Future research should focus on exploring cultural differences in grief, particularly how various cultures respond to loss during crisis situations and the effects of public health crises on mourning practices and rituals. Additionally, it is vital to assess the effectiveness of psychological interventions, such as group therapy and community support, in post-pandemic contexts, specifically examining how these interventions influence grief adaptation and emotional resilience. Moreover, analyzing the experiences of different demographic groups—such as age, socioeconomic status, and ethnicity—during health crises can uncover important nuances that can facilitate the development of tailored support interventions. Long-term studies examining the effects of grief initiated in the context of public health crises are essential for understanding how grief evolves over time, particularly regarding the persistence of grief symptoms and the coping strategies individuals employ.

Comments 12: There are minor inconsistencies in referencing styles. Ensure all citations are formatted uniformly according to journal guidelines.

Response 12: We appreciate your feedback regarding the referencing styles. We have reviewed all citations and ensured that they are formatted uniformly according to the journal guidelines. Thank you for bringing this to our attention.

Reviewer 2 Report

Comments and Suggestions for Authors

Dear authors, I really appreciated your revision work based on the given suggestions and found each entry relevant. Concerns still remain about the excessive textual extension of the introduction and results, which should be helped with schematic tables, as well as the conclusions should be more linear and essential, leaving the rest of the content in the discussions. References should be edited according to editorial rules, and I invite you to reevaluate the corrections. For the effort, despite the limitations of the study, I still suggest minor revisions to further refine it. Good work!

Author Response

Comments 1: Dear authors, I really appreciated your revision work based on the given suggestions and found each entry relevant.

Response 1: Thank you for your thoughtful feedback. We greatly appreciate your recognition of our revision efforts and are glad to hear that you found our responses relevant. Your insights have been invaluable in enhancing the quality of our work.

Comments 2: Concerns still remain about the excessive textual extension of the introduction and results, which should be helped with schematic tables,

Response 2: Thank you for your valuable feedback regarding the excessive textual extension of the introduction. In consideration of one of the reviewers' suggestions, we have reduced the emphasis on certain political aspects that appeared to be excessive in the introduction. We believe this revision streamlines the content while maintaining the necessary context.

The following paragraph was removed:

Thus, grief in Brazil during the pandemic was deeply intertwined with political and social issues that intensified the suffering of the bereaved. Political polarization divided the country and created rifts within families, making it difficult to provide the necessary social support during the mourning process. Additionally, inadequate access to healthcare systems and the economic crisis further deepened inequalities, heightening the sense of vulnerability.

We understand your concerns regarding the textual extension of the results sections. In response, we have created schematic tables to present the results more concisely and enhance clarity.

Comments 3: As well as the conclusions should be more linear and essential, leaving the rest of the content in the discussions.

Response 3: The conclusions have been summarized to focus on the main findings, research limitations, contributions of the study, and suggestions for future investigations. This approach ensures a more linear and essential presentation while allowing the rest of the content to be discussed in detail.

Comments 4: References should be edited according to editorial rules, and I invite you to reevaluate the corrections.

Response 4: We appreciate your feedback regarding the referencing styles. We have reviewed all citations and ensured that they are formatted uniformly according to the journal guidelines. Thank you for bringing this to our attention.

Comments 5: For the effort, despite the limitations of the study, I still suggest minor revisions to further refine it. Good work!

Response 5: Thank you for your thoughtful feedback and encouragement. We appreciate your acknowledgment of our efforts and your suggestions for minor revisions. We are committed to further refining the study and will take your recommendations into account to enhance its quality.